# *Actinobacillus pleuropneumoniae* Eradication with Enrofloxacin May Lead to Dissemination and Long-Term Persistence of Quinolone Resistant *Escherichia coli* in Pig Herds

**DOI:** 10.3390/antibiotics9120910

**Published:** 2020-12-15

**Authors:** Håkon Kaspersen, Anne Margrete Urdahl, Carl Andreas Grøntvedt, Stine Margrethe Gulliksen, Bereket Tesfamichael, Jannice Schau Slettemeås, Madelaine Norström, Camilla Sekse

**Affiliations:** 1Norwegian Veterinary Institute, P.O. Box 750 Sentrum, 0106 Oslo, Norway; Hakon.Kaspersen@vetinst.no (H.K.); anne-margrete.urdahl@vetinst.no (A.M.U.); carl-andreas.grontvedt@vetinst.no (C.A.G.); bereket.tesfamichael@vetinst.no (B.T.); jannice.schau-slettemeas@vetinst.no (J.S.S.); madelaine.norstrom@vetinst.no (M.N.); 2Animalia, P.O. Box 396 Økern, 0513 Oslo, Norway; stine.gulliksen@animalia.no

**Keywords:** *E. coli*, QREC, resistance, quinolone, pig

## Abstract

Norway has a favourable situation with regard to health status and antimicrobial usage in the pig production sector. However, one of the major disease-causing agents in the commercial pig population is *Actinobacillus pleuropneumoniae* (APP). In some herds, APP eradication has been performed by using enrofloxacin in combination with a partial herd depopulation. The aim of this study was to investigate the long-term effects of a single treatment event with enrofloxacin on the occurrence of quinolone resistant *Escherichia coli* (QREC). The study was designed as a retrospective case/control study, where the herds were selected based on treatment history. Faecal samples were taken from sows, gilts, fattening pigs and weaners for all herds where available. A semi-quantitative culturing method was used to identify the relative quantity of QREC in the faecal samples. A significant difference in overall occurrence and relative quantity of QREC was identified between the case and control herds, as well as between each animal age group within the case/control groups. The results indicate that a single treatment event with enrofloxacin significantly increased the occurrence of QREC in the herd, even years after treatment and with no subsequent exposure to quinolones.

## 1. Introduction

Quinolones are categorized as a highest priority drug on the WHO list of critically important antibiotics for human medicine [1], and are restricted for use in animals by the EU Antimicrobial Advice ad hoc Expert Group (AMEG) [2]. Since quinolones are highly mutagenic, the use of such compounds has been linked to an increased occurrence of quinolone resistant *E. coli* (QREC) [3]. However, little is known about the long-term persistence of QREC among pig herds, as most studies have only investigated the persistence of QREC a few months after treatment [4,5]. In Norway, antibiotics are not allowed to be used as growth promoters (feed additives) or for routine prevention of infection in any animal production, and the use of antibiotics in pig production is very low [6]. Less than 1% of the antibiotics used for pigs are quinolones [6], and quinolone resistance among *E. coli* has only been detected at low levels [7].

Norway has a pig production of approximately 1.7 million pigs slaughtered annually for domestic consumption, and negligible import of live pigs from other countries [8,9]. The population is structured in a hierarchical pyramid, with genetic nucleus and multiplier herds comprising the two apex tiers and unidirectional live animal trade downwards to commercial herds with sows and specialized finishing herds. The national pig health status is favourable, as the population is free from several important (viral and bacterial) pathogens that are commonly occurring in many pig populations worldwide [10,11]. However, one of the major respiratory disease causing pathogens in Norway is *Actinobacillus pleuropneumoniae* (APP) [10]. With the exception of specific pathogen-free (SPF) herds that are clinically and serologically negative for APP, this agent is widely prevalent and disease caused by APP is common [10,12,13]. Prevention of clinical disease with APP is largely based on vaccination and optimizing management, but the eradication of the agent from infected herds in Norway relies either on complete depopulation and restocking with replacement animals from APP-free herds, or partial depopulation with strategic antibiotic medication. The latter alternative is generally less costly and preserves the breeding stock, but is less likely to be successful and requires medication of all remaining animals with antibacterial pharmaceuticals.

In the Norwegian pig production sector, enrofloxacin has been used sporadically to medically eradicate APP from a limited number of pig herds [14]. APP eradication with enrofloxacin may be an attractive option due to lower costs and preservation of breeding stock when compared to complete depopulation. However, maintaining the overall low level of QREC in Norwegian pig herds is of interest as well. Thereby, it is important to understand how APP eradication with enrofloxacin may affect the occurrence in exposed herds over time. The aim of this study was to investigate whether/how a single APP eradication event with enrofloxacin impacted the long-term persistence of QREC in Norwegian pig herds.

## 2. Results

In this study, 363 samples were collected from five case herds and 368 samples were taken from five control herds, resulting in 731 samples (Table 1). QREC was detected in 338 of these samples, of which 254 were from case herds and 84 from control herds. The total occurrence of QREC among the case and control herds were 70.0% and 22.8%, respectively. The occurrence of QREC among the samples from case herds were significantly higher than the occurrence among the samples from control herds; χ^2^(1, *N* = 731) = 163.4, *p* < 0.01 (Figure 1).

For all ten farms, 203 samples were obtained from weaners, 140 samples from fattening pigs, 189 samples from gilts, and 199 samples from sows. The occurrence of QREC among pigs sampled in the case herds were 72.0% for weaners, 41.7% for fattening pigs, 80.2% for gilts, and 74.5% for sows. Respectively, the occurrence among pigs sampled in the control herds were 2.9% for weaners, 8.8% for fattening pigs, 29.5% for gilts, and 49.5% for sows (Figure 2). A significantly higher QREC occurrence was observed for all four age groups in the case herds when compared to the same age groups in the control herds (*p* < 0.01) (see Appendix A). In the control herds, an increasing occurrence of QREC was observed with increasing age of the animal, regardless of the herd of origin (Figure 2).

The relative fraction of QREC among the QREC positive case samples was significantly higher compared to the QREC positive control samples *t*(148.2) = 8.95, *p* < 0.01. The amount of QREC compared to total *E. coli* never exceeded 1% among the control samples. Among the case samples, 13 samples had a QREC fraction of 100%, where nine were from weaners and four were from sows (Figure 2).

No ciprofloxacin MIC values above the clinical breakpoint at 0.5 mg/L were observed among the 338 QREC isolates. The ciprofloxacin MIC values ranged from 0.12 to 0.5 mg/L, where the majority of isolates had an MIC value of 0.25 mg/L (Table 2). As for nalidixic acid, the majority of the QREC isolates (*n* = 262, 77.5%) had an MIC value above 64 mg/L, while ten isolates were found to be susceptible (2.95%).

Resistance towards other antimicrobial families in addition to quinolones was observed in 130 (38.5%) of the 338 confirmed QREC isolates. In addition to quinolones, 70 of the 130 isolates (53.8%) were resistant to one additional antimicrobial, 40 (30.8%) were resistant to two additional antimicrobials, and 11 (8.5%) were resistant to three additional antimicrobials (Appendix A). Only nine isolates (6.9%) were resistant to four, five, or six additional antimicrobials. Among the 338 QREC isolates, additional resistance toward tetracycline was observed in 108 isolates (32.0%), while resistance toward sulfamethoxazole was observed in 60 QREC isolates (17.8%). 

Non-metric multidimensional scaling (NMDS) analysis of resistance patterns among the 338 QREC isolates revealed three major clusters containing isolates from several herds, cluster 1, 2, and 3 (Figure 3). Cluster 1 represented isolates with resistance toward ciprofloxacin and nalidixic acid. Cluster 2 represented isolates with resistance toward tetracycline in addition to ciprofloxacin and nalidixic acid. Cluster 3 only contained isolates from control herd one and expressed resistance toward sulfamethoxazole and tetracycline in addition to ciprofloxacin and nalidixic acid. The smaller clusters represented isolates with various resistance patterns.

## 3. Discussion

To our knowledge, this is the first study where the long-term effects of a single quinolone treatment event in pig herds is investigated. The results indicate that the use of enrofloxacin to medically eradicate APP from pig herds likely induces or selects for quinolone resistance in the host’s microbiota. These results are in agreement with other studies that have investigated the short-term effects of quinolone use on the occurrence of QREC and quinolone resistant *Campylobacter* [4,5,15]. Additionally, the results indicate transmission of QREC between the different animal age groups. Taking the year of APP eradication into account, the results also suggest that QREC seems to persist in the farm for a long time, either by persistence of QREC in treated sows and offspring, or persistence in the production environment.

A significantly higher occurrence of QREC was detected among all four age groups in the case herds compared to the control herds. This indicates that QREC has been disseminated between the different age groups, likely from sows, either through direct transfer to offspring, direct contact or from shedding to the environment. Transmission of quinolone resistant *Campylobacter* from fluoroquinolone treated to untreated pigs after only a few days in the same pen has previously been described [15]. In the mentioned study, quinolone resistant *Campylobacter* were detected from all environmental samples taken from the ground, the feed, and the drinking water, presumably through faecal contamination, already after the first day of mixing the treated and untreated herds. This suggests that the quinolone resistant bacteria in the animals are rapidly disseminated to the local environment, which may explain the high occurrence among all the age groups in the present study. Furthermore, a previous study detected similar QREC strains in both treated and untreated groups of pigs, both housed in the same barn, suggesting dissemination between the groups from the environment [4]. These findings strengthen the hypothesis that QREC is disseminated from the originally treated sows to the environment and/or pigs in direct contact. Due to the time elapsed from medication, none of the treated sows were present in the case herds at the time of sample collection. It is therefore probable that QREC persists in the farm and/or in the animals present on the farm.

A significantly higher relative fraction of QREC was detected among the samples from case herds compared to the control herds. This indicates that the amount of QREC in animals from untreated herds may be lower than in the animals from treated herds. However, it should be mentioned that the method used for quantification of QREC and total *E. coli* did not take into account the exact amount of *E. coli* colonies for each dilution. Therefore, in those cases where the relative fraction of QREC and total *E. coli* were the same, it is unknown if the total *E. coli* population is indeed represented as only QREC isolates.

All case herds have bought recruitment gilts after completing the eradications. Unfortunately, as had four out of five of the control herds during the same period [16]. The control herds bought between twelve and 317 pigs during the study period, while the case herds bought between 38 and 332. The use of quinolones in all herds supplying replacement gilts was investigated, and none of these herds had used quinolones [17].

The NMDS analysis on resistance patterns revealed a cluster consisting of isolates from only one control herd (Figure 3). This may indicate that the QREC isolates detected in this herd may be highly similar, and may have been disseminated between the different age groups to a higher degree when compared to the other herds. Case herd D is represented in three different clusters, which indicates a larger diversity of QREC in that herd. A previous study found that the diversity of *E. coli* varied greatly within each pig in both the treated and untreated groups, even many days after treatment [4]. Since the current study only investigated the occurrence of QREC at one point in time, it is impossible to determine if the same QREC strain persisted within each herd. A longitudinal study, including population genomics, would be needed to investigate these findings further.

The low MIC values for ciprofloxacin among the QREC isolates in this study indicate that few quinolone resistance mechanisms coincide within the isolates, as higher MIC values are typically observed in isolates with multiple quinolone resistance mechanisms [18]. Reduced susceptibility to ciprofloxacin (MIC > 0.06 mg/L) in combination with susceptibility towards nalidixic acid has previously been found to indicate the presence of plasmid mediated quinolone resistance (PMQR) mechanisms [19]. In addition, PMQR has been identified in 26.6% of the included samples from fattening pigs in a previous study in Norway [20]. In the present study, the majority of the QREC isolates had a relatively high resistance level toward nalidixic acid (128–256 mg/L). This indicates a low occurrence of PMQR among the studied pig populations, in contrast to the relatively high occurrence reported in the previous study. However, whole genome sequencing and in-silico detection of these genes are needed to verify these findings.

To summarise, the results indicate dissemination of QREC from the originally treated sows into the environment and/or offspring, and that these bacteria persist even years after the treatment event. 

## 4. Materials and Methods 

### 4.1. Study Design

This study was designed as a case-control study. Sow herds that had previously been subject to partial depopulation and APP eradication with enrofloxacin by the Norwegian pig production sector were included as case herds. In short, in the case herds, all animals of <10 months of age had been removed prior to medication. All remaining animals were treated two to three times with 1.5 mL/20 kg (7.5 mg/kg) enrofloxacin given as an intramuscular injection, with an interval of three days between injections. Additionally, all units and pens were cleaned and disinfected. Cleaning and disinfection were preferably performed in empty units. The herds were restocked mainly by breeding the medicated sows, and to a lesser extent by purchasing recruitment gilts from supplying herds.

For the control herds, two criteria had to be fulfilled: (i) the herd had, with a high certainty, not been treated with quinolones, and (ii) the case and control herds were serviced by the same veterinary practitioner. In addition, the herd had preferably been closed the last 10 years.

For each herd, up to 20 faecal samples were collected from weaners, fattening pigs, gilts, and sows, when available. Not all age groups were represented within each farm. Sampling was performed during 2017–2019. Samples were sent by post over night from the farm to the laboratory at the Norwegian Veterinary Institute and the fresh faecal samples were analysed the day they arrived.

### 4.2. Laboratory Methods

A semi-quantitative method, as previously described [21], was used to determine the relative amount of QREC present in each sample compared to the total amount of *E. coli*. A 1:10 initial dilution was made by mixing 1 g of faeces with 9 mL of buffered peptone water (BPW) for each sample. Ten-fold dilutions were made from this suspension by mixing 20 µL with 180 µL 0.9% saline solution on a microtiter plate. For each dilution, 20 µL of the previous dilution was used to create the next dilution, to a final dilution of 1:10^6^. Each dilution was mixed thoroughly by pipetting. Then, 10 µL from each dilution, including the first 1:10 dilution (in total six), were plated onto two square MacConkey agar plates with and without 0.06 mg/L ciprofloxacin. The plating was done at an approximately 45 degree angle, letting the suspension flow to 1–2 cm from the bottom of the plate. The suspension streaks were air-dried, and the plates were incubated at 41.5 °C for 18–22 h. The initial faecal suspensions were incubated under the same conditions as above.

Following incubation, *E. coli* colonies were selected based on morphology. Putative *E. coli* were confirmed by using a matrix-associated laser desorption ionization time-of-flight (MALDI-TOF Microflex, Bruker Daltonik GmbH) instrument. After being confirmed as *E. coli*, growth from each dilution was registered as present (1) or absent (0) from both MacConkey agar plates with and without 0.06 mg/L ciprofloxacin. If present, up to three presumptive QREC isolates per sample were plated onto MacConkey with 0.06 mg/L ciprofloxacin or blood agar. These isolates were further susceptibility tested using broth microdilution (EUVSEC1, Sensititre, ThermoFisher Scientific) to determine the minimum inhibitory concentration (MIC) values towards 13 different antimicrobials. If no growth of *E. coli* was detected on the MacConkey agar with 0.06 mg/L ciprofloxacin, 10 µL of the initial 1:10 dilution was plated onto only MacConkey with 0.06 mg/L ciprofloxacin, followed by incubation at 41.5 °C for 18–22 h. Presumptive *E. coli* on these plates were handled in the same manner as described above. One *E. coli* from MacConkey without 0.06 mg/L ciprofloxacin per sample was plated onto blood agar and further stored at −80 °C.

A sample was regarded as QREC positive if the MALDI-TOF confirmed one isolate from the sample as *E. coli,* and the MIC value for ciprofloxacin of this isolate was above 0.06 mg/L, based on the epidemiological cut-off value defined by the European Committee on Antimicrobial Susceptibility Testing (EUCAST, retrieved 15.05.2019). The total amount of *E. coli* in each sample was calculated based on the presence of *E. coli* colonies at the respective dilutions (Appendix A). The relative percentage of QREC in relation to total *E. coli* was calculated for each sample.

### 4.3. Statistical Analysis

All statistics and visualizations were done in R [22] version 4.0.0. Significant differences in occurrence of QREC between groups were calculated using χ^2^-tests. A Welch two-sample t-test was used to determine if the relative fraction of QREC was higher in the QREC positive case samples compared to the QREC positive control samples. To be able to use a two-sample t-test, the calculated fractions of QREC within each QREC positive sample were log10-transformed. Then, a variance test was used to investigate if the variances between the two groups were equal. The distribution of the log10-transformed fractions were plotted to see if they were normally distributed (Appendix A).

Non-metric multidimensional scaling (NMDS) was used to cluster QREC isolates based on resistance patterns toward 13 different antimicrobials. First, pairwise jaccard distances were calculated from the presence (1) or absence (0) of resistance for each antimicrobial. Then, these distances were used in an NMDS analysis using the metaMDS function from the R package “vegan” [23], with two dimensions and 200 random starts.

## 5. Conclusions

The results indicate that a single herd-treatment event with enrofloxacin during a very limited period of time significantly increases the occurrence of QREC in the herd, even after several years of no quinolone exposure.

## Figures and Tables

**Figure 1 antibiotics-09-00910-f001:**
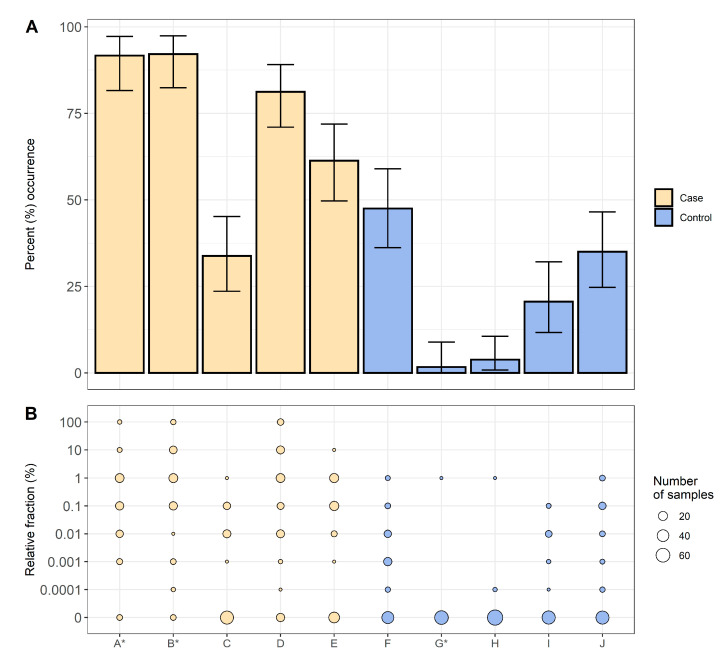
Occurrence and relative fraction of QREC among the tested samples from each herd. Case herds are coloured in yellow, and control herds in blue. Plot (**A**) presents the occurrence of QREC for each herd (x-axis) with 95% confidence intervals. Plot (**B**) presents the relative fraction of QREC among the total *E. coli* detected in the samples. The point size represent the number of samples with the respective QREC fraction. * herd is lacking samples from fattening pigs.

**Figure 2 antibiotics-09-00910-f002:**
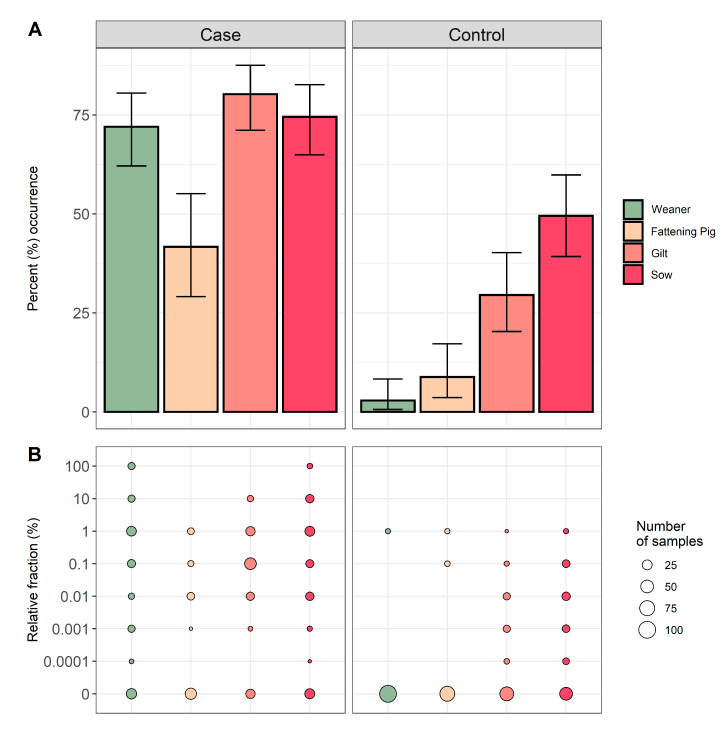
Occurrence and relative fraction of QREC among the tested samples from all four age groups. The colours denote the four different age groups included; weaners (green), fattening pigs (yellow), gilts (light red), and sows (dark red). Plot (**A**) presents the percent occurrence of QREC with 95% confidence intervals. Plot (**B**) represents the relative fraction of QREC among the total *E. coli* in the samples. The point size represents the number of samples with the respective QREC fraction.

**Figure 3 antibiotics-09-00910-f003:**
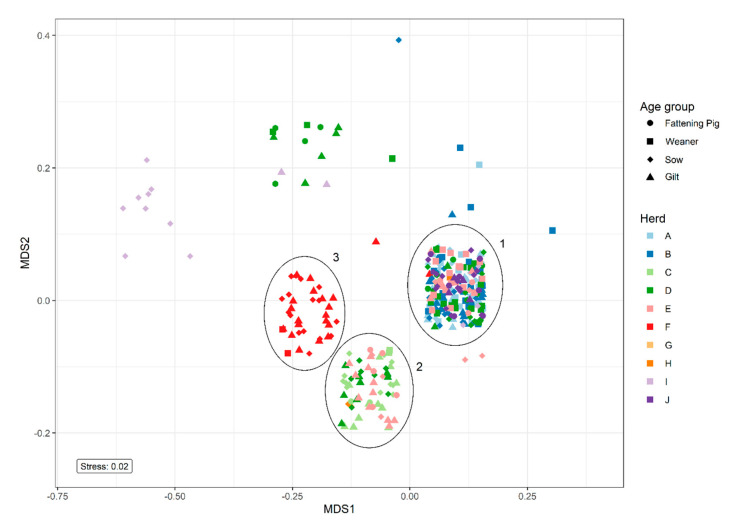
Non-metric multidimensional scaling (NMDS) analysis of resistance patterns among the 338 QREC isolates, colour-separated by herd. The shapes represent the age group of the animal. Pairwise Jaccard distances between isolates were calculated based on the presence (1) or absence (0) of resistance towards 13 different antimicrobials. The NMDS analysis was run with two dimensions and 200 random starts. The points are jittered for easier interpretation. Rings denote major clusters with the following resistance patterns: Cluster 1: ciprofloxacin and nalidixic acid, Cluster 2: tetracycline, ciprofloxacin and nalidixic acid, Cluster 3: sulfamethoxazole, tetracycline, ciprofloxacin, and nalidixic acid.

**Table 1 antibiotics-09-00910-t001:** Number of samples per herd and age group. The “Year” column denotes the year of *Actinobacillus pleuropneumoniae* eradication. Columns denote the number of samples per age group, and rows denote the number of samples per herd. The rows denoted “Sum” show the number of samples in total per age group for case and control herds, respectively.

	Herd	Year	Weaner	Fattening Pig	Gilt	Sow	Sum
Case	A	2016	20	0	20	20	60
	B	2015	20	0	21	22	63
	C	2016	20	20	20	20	80
	D	2009	20	20	20	20	80
	E	2014	20	20	20	20	80
Sum			100	60	101	102	363
Control	F		23	20	20	17	80
	G		20	0	20	20	60
	H		20	20	20	20	80
	I		20	20	8	20	68
	J		20	20	20	20	80
Sum			103	80	88	97	368
Total			203	140	189	199	731

**Table 2 antibiotics-09-00910-t002:** Minimum inhibitory concentration distribution for all quinolone resistant *E. coli* isolates in this study (*n* = 338). Vertical lines denote the epidemiological cut-off (ECOFF) values for each antimicrobial, defined by EUCAST. The abbreviated antimicrobial names are listed in the column “AM”; AMP = ampicillin, CHL = chloramphenicol, CIP = ciprofloxacin, COL = colistin, CTX = cefotaxime, GEN = gentamicin, MEM = meropenem, NAL = nalidixic acid, SMX = sulfamethoxazole, CAZ = ceftazidime, TET = tetracycline, TGC = tigecycline, and TMP = trimethoprim. Percent resistances, based on the ECOFF values, are listed in the column “Percent”, with 95% confidence intervals for each respective antimicrobial. White areas denote the range of concentrations tested for each antimicrobial. Numbers outside this area denote growth on all test concentrations. Azithromycin was excluded from this table as no ECOFF value is currently defined by EUCAST for this antimicrobial.

AM.	Percent	95% CI	0.015	0.03	0.06	0.12	0.25	0.5	1	2	4	8	16	32	64	128	256	512	1024	>1024
AMP	6.8	[4.4–10]							29	93	137	56	1			22				
CHL	2.4	[1.0–4.6]										325	5	8						
CIP	100	[98.9–100]				98	164	76												
COL	0	[0.0–1.1]							332	6										
CTX	0.3	[0.0–1.6]					337					1								
GEN	3.8	[2.1–6.5]						160	144	21	3	1		8	1					
MEM	0	[0.0–1.1]		338																
NAL	97	[94.6–98.6]									4	6	1	14	51	136	126			
SMX	17.8	[13.8–22.3]										264	13	1			1	2	26	31
CAZ	0.3	[0.0–1.6]						337					1							
TET	32	[27.0–37.2]								221	9			9	80	19				
TGC	0.9	[0.2–2.6]					299	36	3											
TMP	4.7	[2.7– 7.6]					282	36	2	2	3	1			12

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
