# Peer review of "Actinobacillus pleuropneumoniae Eradication with Enrofloxacin May Lead to Dissemination and Long-Term Persistence of Quinolone Resistant Escherichia coli in Pig Herds"

_antibiotics, 2020, doi:10.3390/antibiotics9120910_

Round 1

Reviewer 1 Report

The manuscript explored quinolones resistance in pig farms. The approach is both empirical and practical/applicable to farms. It is excellently written with a few clarifications needed. The attached manuscript contains areas that need to be addressed. 
I will pass the manuscript for publication in an international journal like Antibiotics.

Author Response

Reply to Reviewer 1

  1. Is there a reference for this observation or it is based on the knowledge of the field?

L45-47: With the exception of specific pathogen-free (SPF) herds that are clinically and serologically negative for APP, this agent is widely prevalent and disease caused by APP is common.

Answer: Thanks for pointing out the lack of references here. We have added one reference for the international situation and two references for the Norwegian situation as no single reference covers the Norwegian situation. Unfortunately, one of the references regarding the Norwegian situation is a report in Norwegian.

  1. If the Chi square value is 163.4, it should be placed directly after Chi square.

L71-73: The occurrence of QREC among the samples from case herds were significantly higher than the occurrence among the samples from control herds; χ2(1, N = 731) = 163.4, p < 0.01 (Figure 1).

Answer: We have followed the APA guidelines for reporting Chi-square test results. If this is something that the Journal wishes to be changed, we can accommodate this.

  1. What is the Chi Square value?

L90-91: In the control herds, an increasing occurrence of QREC was observed with increasing age of the animal, regardless of the herd of origin (Figure 2).

Answer: Since this was a result of four different chi-square tests, we did not report the values in the text. However, these values can now be found in a new table, supplementary table 1, where all the results of these four tests are reported.

  1. You cannot state this in third person since you are the authors.

L142-143: To the authors’ knowledge, this is the first study where the long-term effects of a single quinolone treatment event in pig herds is investigated.

Answer: We have changes this to “our” as suggested by the reviewer.

Reviewer 2 Report

General Comments

The emergence of resistant bacterial pathogens in animal productions to Highest Priority Critically Important Antimicrobials (HPCIAs) for human medicine, such as (fluoro)quinolones, represents an increasing threat for several countries worldwide. Maintaining an overall low level of AMR in those few countries with a current favourable situation on antimicrobial usage and resistance (such as Norway) is of great relevance. In this study (designed as case-control), the Authors aimed to investigate the long-term effects of a single treatment for pleuropneumonia in pigs with enrofloxacin on the occurrence of quinolone resistant Escherichia coli (QREC). Unfortunately, the study is rather limited and the lack of molecular insights into the genetic basis of QREC does not allow to verify and properly discuss the main results obtained.

An additional weakness of this study (as also underlined by the Authors in the discussion section), is that the persistence of QREC at herd level was not investigated by further longitudinal analysis and genetic characterization/typing of QREC conducted within one or more herds. In my opinion, this represents a fundamental component of an investigation on the long-term persistence of QREC in pig herds, where the persistence of certain clones or plasmid-related transmission would determine the persistence of resistant commensal E.coli.

Scientific novelty: To date, information on the long-term effects of a single treatment with (fluoro)quinolones in animal productions (in terms of increasing commensal QREC), is still limited and this study would aim to address it.

-Writing style and English language: In general, the writing style of the text is very concise and informative.

Specific Comments

The manuscript would also benefit from the following specific modifications:

Title: I would suggest to change the title as follows:

Actinobacillus pleuropneumoniae eradication with Enrofloxacin May Lead to Dissemination and Long-Term Persistence of Quinolone Resistant Escherichia Coli in Pig Herds”

In particular I would suggest to replace the sentence “medical eradication” here and throughout the text with “APP eradication”.

Abstract

Lines 14-15 please rephrase as follows: “APP eradication has been sporadically achieved by using enrofloxacin in infected herds in combination with a partial herd depopulation”.

Results

Table 1 Please replace “sum” with “total”

Lines 104-108 “No ciprofloxacin MIC…susceptible”. Molecular investigation of the genetic basis (as presence of chromosomal point mutations in gyrA/parC genes, mutations on porins and multidrug efflux pumps, PMQR genes), would have been necessary to possibly explain the obtained MIC values and related (fluoro)quinolone resistance phenotypes

Line 110 Table caption: could the Authors clarify what do they refer to vertical lines? It seems that there are not vertical lines in Table 1. Do the Authors refer to the gray part for MIC values below the ECOFFs? Is it possible?

Lines 117-130 Similarly to previous comment (Lines 104-108), molecular investigation of the genetic basis of resistance patterns to the other antimicrobial classes other than (fluoro)quinolone, would have been necessary to support the MDR resistance phenotypes and related clusterization revealed by NMDS analysis.

Discussion

Lines 185-187: As reported in General Comments, this point represents a major weakness.

Lines 192-198: As reported in General and Results Comments, this point represents a major weakness. Without further molecular characterization, the obtained MIC values can not be evaluated.

Lines 203-205 Dosage regimen is lacking. It shoud be specified in the M&M section. If it refers to what has been reported in the Introduction section (Lines 56-57), this part should be moved to the M&M section.

Lines 209-211 Timeframe between the start of treatment and the time of sampling is not clear. It should be mentioned here in the M&M section. Were all the enrolled herds tested before starting the study to see if they were already positive for QREC?  Were all the samples analyzed in the same lab using the same protocols? More information on single herds (e.g. animals turnover and treatment protocols administered) should be given.

Lines 241-242 Is it correct referring to Table 1? Where did the Authors report the total amount of E.coli?

Author Response

Please find the reply to reviewer 2 in the attached file

Round 2

Reviewer 2 Report

I have no further comments on the version 2 of the document.